# Performance evaluation of the Ortho VITROS SARS-CoV-2 Spike-Specific Quantitative IgG test by comparison with the surrogate virus neutralizing antibody test and clinical assessment

Maika Takahashi[1], Kaori Saito[2], Tomohiko Ai[2], Shuko Nojiri[3], Abdullah Khasawneh[2], Faith Jessica Paran[4], Yuki Horiuchi[2], Satomi Takei[2], Takamasa Yamamoto[1], Mitsuru Wakita[1], Makoto Hiki[5,6], Takashi Miida[2], Toshio Naito[4,7], Kazuhisa Takahashi[4,8], Yoko Tabe[2,4]*

1 Department of Clinical Laboratory, Juntendo University Hospital, Bunkyo City, Tokyo, Japan, 2 Department of Clinical Laboratory Medicine, Juntendo University Graduate School of Medicine, Bunkyo City, Tokyo, Japan, 3 Medical Technology Innovation Center, Juntendo University, Bunkyo City, Tokyo, Japan, 4 Department of Research Support Utilizing Bioresource Bank, Juntendo University Graduate School of Medicine, Bunkyo City, Tokyo, Japan, 5 Department of Emergency Medicine, Juntendo University Faculty of Medicine, Bunkyo City, Tokyo, Japan, 6 Department of Cardiovascular Biology and Medicine, Juntendo University Faculty of Medicine, Bunkyo City, Tokyo, Japan, 7 Department of General Medicine, Juntendo University Graduate School of Medicine, Bunkyo City, Tokyo, Japan, 8 Department of Respiratory Medicine, Juntendo University Graduate School of Medicine, Bunkyo City, Tokyo, Japan

* tabe@juntendo.ac.jp

## Abstract

### Background

Despite the worldwide campaigns of COVID-19 vaccinations, the pandemic is still a major medical and social problem. The Ortho VITROS SARS-CoV-2 spike-specific quantitative IgG (VITROS S-IgG) assay has been developed to assess neutralizing antibody (NT antibody) against SARS-CoV-2 spike (S) antibodies. However, it has not been evaluated in Japan, where the total cases and death toll are lower than the rest of the world.

### Methods

The clinical performance of VITROS S-IgG was evaluated by comparing with the NT antibody levels measured by the surrogate virus neutralizing antibody test (sVNT). A total of 332 serum samples from 188 individuals were used. Of these, 219 samples were from 75 COVID-19 patients: 96 samples from 20 severe/critical cases (Group S), and 123 samples from 55 mild/moderate cases (Group M). The remaining 113 samples were from 113 healthcare workers who had received 2 doses of the BNT162b2 vaccine.

### Results

VITROS S-IgG showed good correlation with the cPass sVNT assay (Spearman rho = 0.91). Both VITROS S-IgG and cPass sVNT showed significantly higher plateau levels of

**Data Availability Statement:** All relevant data are presented and shared in the main figures and tables of the paper.

**Funding:** This work was supported in part by Japan Agency for Medical Research and Development (grant No. JP20fk0108472) to Toshio Naito and by Japan Society for the Promotion of Science Grants-in Aid for Scientific Research (grant No. 22K15675) to Dr. Satomi Takei. The funders had no role in study design, data collection and analysis, decision to publish, or preparation of the manuscript.

**Competing interests:** Ortho Clinical Diagnostics provided Anti-SARS-CoV-2 IgG Quantitative Reagent, and Roche Diagnosis provided reagents for Elecsys Anti-SARS-CoV-2 assay free of cost to the researchers. The companies did not take part in 1) the study design, 2) the data interpretation, and 3) the writing of this paper. This does not alter our adherence to PLOS ONE policies on sharing data and materials.

antibodies in Group S compared to Group M. Regarding the humoral immune responses after BNT162b2 vaccination, individuals who were negative for SARS-CoV-2 nucleocapsid (N)-specific antibodies had statistically lower titers of both S-IgG and sVNT compared to individuals with a history of COVID-19 and individuals who were positive for N-specific antibodies without history of COVID-19. In individuals who were positive for N-specific antibodies, S-IgG and sVNT titers were similar to individuals with a history of COVID-19.

## Conclusions

Although the automated quantitative immunoassay VITROS S-IgG showed a reasonable correlation with sVNT antibodies, there is some discrepancy between Vitros S-IgG and cPass sVNT in milder cases. Thus, VITROS S-IgG can be a useful diagnostic tool in assessing the immune responses to vaccination and herd immunity. However, careful analysis is necessary to interpret the results.

## Introduction

Coronavirus disease 19 (COVID-19), caused by SARS-CoV-2 infection, is an unprecedented threat to public health and the economy [1]. The absence of specific treatment options has resulted in the important implementation of precautions and diagnostic testing.

The usual choice for COVID-19 diagnosis is molecular testing, particularly RT-PCR, which is a reliable tool for detecting active SARS-CoV-2 infection [2]. Antigen testing has been also developed for rapid detection of pathogens without complicated procedures [3]. However, these tests cannot detect SARS-CoV-2 during certain periods after infection [4]. PCR and antigen tests for virus detection are not competing options for exposure detection, since they can be performed at different time points within their relevant diagnostic windows of clinical development [5]. Therefore, serological tests detecting SARS-CoV-2-specific antibodies have been used as a complement to RT-PCR and antigen testing in the diagnosis of COVID-19 [6–8].

Furthermore, serological tests are essential tools to evaluate neutralizing antibody (NT antibody) titers upon vaccination and to assess SARS-CoV-2 seroprevalence in cohorts [9, 10]. NT antibodies targeting the receptor-binding domain (RBD) of the spike (S) protein can reduce viral infectivity by binding to the surface epitopes of viral particles, blocking virus entry into host cells [11]. Therefore, there is a need for a widely available assay that correlates well with neutralizing activity, has a short turnaround time, has high throughput, and is cost effective.

SARS-CoV-2 serologic assays using spike proteins as target antigens are known to be correlated with virus neutralization activity [12], which can be a pivotal tool for assessing the effect of vaccination. VITROS Immunodiagnostic Products Anti-SARS-CoV-2 IgG Quantitative Reagent (VITROS S-IgG), released by Ortho Clinical Diagnostics, was developed for the detection of IgG antibodies against the S1 subunit including receptor binding domain (RBD) of the spike protein of SARS-CoV-2.

GenScript cPass SARS-CoV-2 Neutralization Antibody Detection Kit, an enzyme-linked immunosorbent assay (ELISA)-based surrogate virus neutralization test (sVNT), mimics the reaction between human ACE2 receptor and RBD. It has been reported that the cPass SARS-CoV-2 NT antibody test (cPass sVNT) is a useful indicator of virus-neutralizing activity and has a good correlation with the cell-culture-based virus neutralization assay using live SARS-CoV-2, the gold standard method of assessing NT antibodies [13, 14].

To evaluate the clinical performance of VITROS S-IgG in detecting neutralizing activity, we investigated the quantitative correlation between it and the cPass sVNT.

## Materials and methods

### Patient cohorts

A total of 188 individual (332 samples) were included in this study. This includes 219 samples obtained from 75 laboratory-confirmed COVID-19 cases between April 2020 and January 2021, and 113 samples from 113 healthcare workers 2 months after their second doses of BNT162b2 vaccine between March and April 2021 in the Juntendo University Hospital, located in Tokyo, Japan. All samples were obtained from Juntendo University Hospital in Tokyo, Japan. A confirmed case of COVID-19 was defined as a positive result of a RT-PCR assay from pharyngeal swab specimens using the 2019 Novel Coronavirus Detection Kit (Shimadzu, Kyoto, Japan). We first categorized SARS-CoV-2 infected patients into mild, moderate, severe, and critical according to the WHO criteria (https://www.who.int/publications/i/item/WHO-2019-nCoV-clinical-2021-2). Mild COVID-19 was defined as respiratory symptoms without evidence of pneumonia or hypoxia, while moderate or severe infection was defined as presence of clinical and radiological evidence of pneumonia. In moderate cases, $SpO_2 \geq 94\%$ is observed in room air, while one of the following was required to identify the severe and critical cases: respiratory rate >30 breaths/min or $SpO_2$ <94% on room air. Critical illness was defined as respiratory failure, septic shock, and/or multiple organ dysfunction (COVID-19 Clinical management: living guidance https://www.who.int/publications/i/item/clinical-management-of-covid-19). We then organized them into Group M, which included mild and moderate cases, and Group S, which included severe and critical cases. Group M patients with a high-risk background were hospitalized and included in the long-term evaluation study.

Of the 75 confirmed COVID-19 patients, 20 cases fall under Group S (critical 4, severe 16) and produced 96 samples (critical 25, severe 71), while 55 cases (moderate 51, mild 4) fall under Group M and produced 123 samples (moderate 103, mild 20).

This study was approved by the Juntendo University Hospital institutional review board (IRB # 20–036) and conducted according to the Helsinki Declarations, using the opt-out method of the hospital website.

### Serologic testing for SARS-CoV-2 by Ortho VITROS SARS-CoV-2 Spike-Specific Quantitative IgG (VITROS S-IgG)

The IgG antibodies against the S1 subunit of the spike protein of SARS-CoV-2 were quantitatively measured using VITROS Immunodiagnostic Products Anti-SARS-CoV-2 IgG Quantitative Reagent (Ortho Clinical Diagnostics, New Jersey) on the VITROS 3600 automated immunoassay analyzer (Ortho Clinical Diagnostics). The VITROS Anti-SARS-CoV-2 IgG assay is a chemiluminescent enzyme immunoassay (CLEIA) using a solid-phase SARS-CoV-2 spike protein antigen to capture antibodies and a horseradish peroxidase (HRP)–labeled recombinant SARS-CoV-2 antigen as a detection reagent. The assay is qualitative, and reports results as reactive or nonreactive based on a manufacturer-defined cutoff index (COI; signal sample/ cutoff) of 1.0, with reactive values falling above this decision limit and nonreactive values below. Placement of the cut-off for a reactive sample is set to ≥17.8 BAU (Binding Antibody Units) /mL (https://www.fda.gov/media/150675/download).

The SARS-CoV-2 nucleocapsid-specific total immunoglobulin (N-total Ig) was measured using Elecsys Anti-SARS-CoV-2 electrochemiluminescence immunoassay (Roche Diagnosis,

Basel, Switzerland) on a cobas e801 analytical unit. The immunoassay utilizes a double-antigen sandwich test principle and a recombinant protein representing the nucleocapsid antigen for the determination of antibodies to SARS-CoV-2. The results are presented in the form of COI. A COI≧1.0 was interpreted as positive. (https://www.fda.gov/media/137605/download).

## Surrogate virus neutralizing antibody detection test by the GenScript cPass SARS-CoV-2 Antibody Detection Kit (cPass sVNT)

Following the company's instructions, surrogate virus neutralizing (sVN) antibodies were measured by the GenScript cPass SARS-CoV-2 Antibody Detection Kit (cPass sVNT), a blocking enzyme-linked immunosorbent assay (GenScript, Piscataway, New Jersey, USA). The samples and controls were briefly pre-incubated with the HRP-labeled recombinant RBD proteins and the mixture was added to the capture plate pre-coated with the hACE2 proteins. After the complex of sVN antibody with RBD-HRP was removed by washing, the wells were read at 450 nm in a microtiter plate reader. The percent signal inhibition for the detection of sVN antibodies were calculated as follows:

% Signal Inhibition = (1—OD value of Sample /OD value of Negative Control) × 100% (cutoff value: 30% signal inhibition). The specifications of VITROS S-IgG and cPass sVNT are summarized in Table 1.

## Statistical analysis

Statistical analyses were performed using GraphPad prism. (GraphPad Software, San Diego, California, USA, www.graphpad.com). Correlation analysis between VITROS S-IgG and cPass sVNT titers was performed using Spearman correlation coefficient. For experiments involving only two groups, the Mann-Whitney U test and Kruskal-Wallis test were performed.

For longitudinal analysis, when experiments involved more than two groups, one-way analysis of variance (ANOVA) followed by Tukey multiple-comparison post hoc analysis were used to analyze statistical differences. Models were fitted to a four-parameter logistic function, with a constrained lower asymptote set to the limit of detection, the infection point, a scale

**Table 1. Specifications of VITROS S-IgG and cPass sVNT.**

| Product Name | VITROS Immunodiagnostic Products Anti-SARS-CoV-2 IgG Quantitative Reagent Pack | cPass™ SARS-CoV-2 Neutralization Antibody Detection Kit |
|---|---|---|
| Manufacturer | Ortho Clinical Diagnostics Inc. | GenScript USA Inc. |
| Platform | VITROS 3600 analyzer | ELISA system |
| Method | CLEIA | ELISA |
| Target antigen | Spike protein S1 | RBD |
| Immunoglobulin class | IgG | Pan-Ig |
| Sensitivity (%, 95% CI) | 91.9 (87.7–95.1)* | 100 (87.1–100.0) ** |
| Specificity (%, 95% CI) | 100 (99.3–100.0) | 100 (95.8–100.0) *** |
| Unit | BAU/mL | % |
| Cut-off | 17.8 | 30 |

CLEIA, chemiluminescent enzyme immunoassay; ELISA, enzyme-linked immunosorbent assay; RBD, receptor binding domain

*Sensitivity was calculated from PCR positive samples collected after 15 days or later from symptom onset.

**Positive percent agreement with plaque reduction neutralization test (PRNT).

***Negative percent agreement with PRNT.

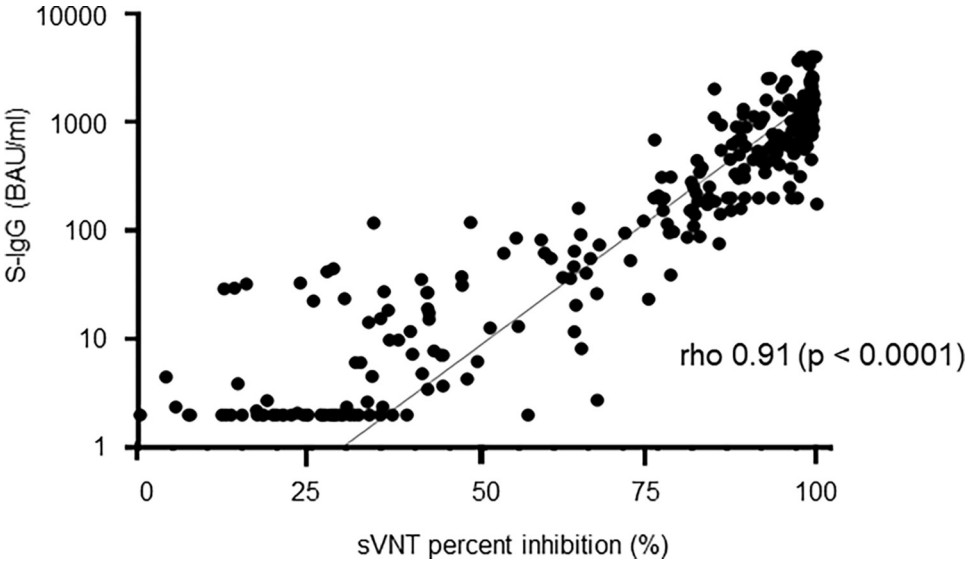

**Fig 1. Comparison of VITROS S-IgG values and cPass sVNT titers.**

parameter, and the upper asymptote for Group S and Group M. sVN antibody titers were fitted and a comparison between Group S and Group M was conducted in a Z test from the estimations.

## Results

### Correlation of VITROS S-IgG and cPass sVNT

Fig 1 shows the correlation between the simultaneously-measured quantitative VITROS S-IgG and cPass sVNT values in 277 samples, including 164 samples from 64 COVID-19 patients and 113 samples from 113 vaccinated individuals. The correlation of the quantitative results of VITORS S-IgG with % inhibition values of cPass sVNT was 0.91 of Spearman's rho value ($p < 0.0001$).

The concordance between the qualitative results of VITROS S-IgG and cPass SARS-CoV-2 neutralization test in SARS CoV-2 positive patients are shown in Table 2. The positive percent agreement of VITROS S-IgG with cPass sVNT was 85.0% (198/233), and the negative percent agreement was 84.1% (37/44). The overall percent agreement of VITROS S-IgG with cPass sVNT was 84.8% (235/277). Among VITROS S-IgG positive samples, 96.6% (198/205) showed

**Table 2. Agreement between VITROS S-IgG and cPass sVNT.**

|  | cPass sVNT | |
| --- | --- | --- |
|  | **positive** | **negative** |
| VITROS S-IgG |  |  |
| positive | 198 | 7 |
| negative | 35 | 37 |

Correlation of VITROS S-IgG and cPass sVNT was evaluated using 277 serum samples: 164 from 64 COVID-19 patients and 113 from 113 vaccinated individuals. P value was evaluated by Spearman's rank-order correlation coefficient (rho). The vertical axis is in logarithmic notation.

Seroprevalence and changes of VITROS S-IgG antibody titers in COVID-19 patients

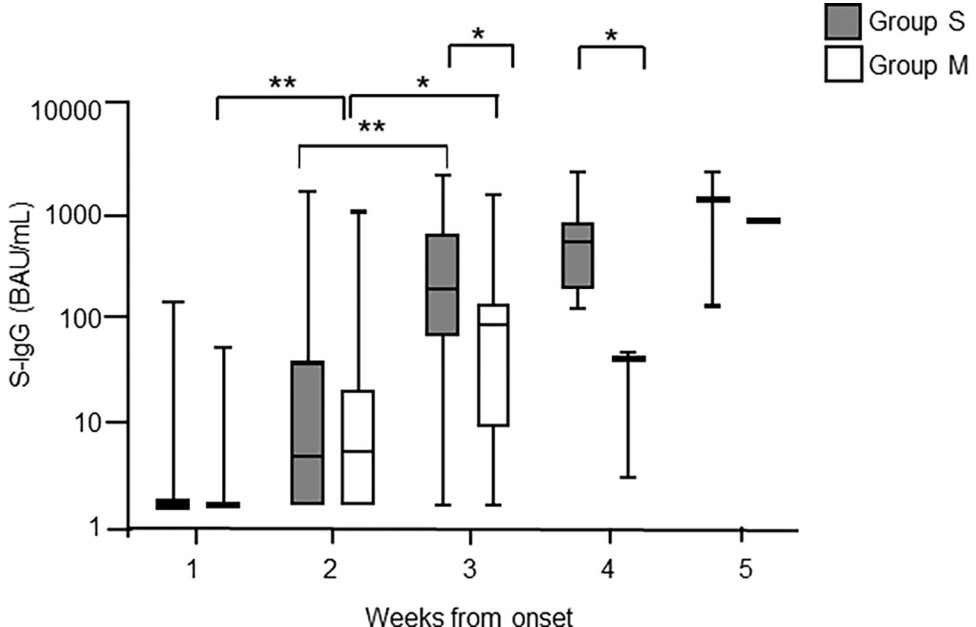

**Fig 2. The time course of VITROS S-IgG titers in COVID-19 patients after symptom onset.**

positive for cPass sVNT. However, we observed that 48.6% of VITROS S-IgG negative samples were cPass sVNT positive (35/72).

The seroprevalence of S-IgG was investigated using the VITROS S-IgG antibody assay with 219 longitudinally assessed samples from the 75 COVID-19 patients. Fig 2 shows chronological changes of S-IgG antibody titers and positivities detected by the VITROS S-IgG antibody assay after symptom onset. In Group S, the S-IgG level increased every week after onset with a significant increase from week 2 to week 3 ($p<0.0001$, Kruskal-Wallis test). In Group M, S-IgG level increased significantly from week 1 to week 2 ($p<0.0001$) and week 2 to week 3 ($p<0.05$). The S-IgG value of Group S was significantly higher than that of Group M during week 3 and week 4 after symptom onset ($p<0.0001$, Mann–Whitney U test). The clinical sensitivity of VITROS S-IgG was shown in Table 3. Sensitivity increased proportionally with time post-infection, reaching approximately 40% in critical, severe, and moderate cases 2 weeks after symptom onset, and 100% after 4 weeks. However, in mild cases, no VITROS S-IgG seroconversion was observed even at 4 weeks after onset.

S-IgG titers were measured for SARS-CoV-2 PCR-positive patient samples for the indicated weekly timeframes post symptom onset using the VITROS S-IgG antibody assay. Ninety-six

**Table 3. Clinical sensitivity of VITROS S-IgG.**

| weeks from onset | critical (n = 4)* | | severe (n = 16) | | moderate (n = 51) | | mild (n = 4) | |
|---|---|---|---|---|---|---|---|---|
| | sample # | positive # (%) | sample # | positive # (%) | sample # | positive # (%) | sample # | positive # (%) |
| 1 | 3 | 0 (0) | 10 | 1 (10) | 36 | 2 (6) | 9 | 0 (0) |
| 2 | 11 | 4 (36) | 36 | 15 (42) | 45 | 17 (38) | 7 | 0 (0) |
| 3 | 8 | 5 (63) | 18 | 18 (100) | 19 | 16 (84) | 3 | 0 (0) |
| 4 | 3 | 3 (100) | 5 | 5 (100) | 2 | 2 (100) | 1 | 0 (0) |
| 5 | 0 | 0 (N/A) | 2 | 2 (100) | 1 | 1 (100) | 0 | 0 (N/A) |

* patient number

N/A, not applicable

samples from 20 severe to critical cases (Group S) and 123 samples from 55 mild to moderate cases (Group M) were tested. The levels of S-IgG antibody in Group S and Group M were compared. Gray bars indicate Group S and open bars indicate Group M. The vertical axis is in logarithmic notation. The data are presented as means with interquartile ranges. Statistical significance is indicated as follows: $^{*}p < 0.05$, $^{**}p < 0.0001$ (Mann-Whitney U test).

## Kinetics of surrogate neutralizing antibody in COVID-19 patients

Next, we evaluated the kinetics of sVN antibody using the cPass sVNT with 164 longitudinally-assessed samples from the 65 COVID-19 patients—84 samples from 20 patients of Group S and 80 samples from 45 of Group M.

Changes of cPass sVNT titers after symptom onset are shown in Fig 3. In Group S, cPass sVNT levels increased every week after symptom onset with a significant increase from week 2 to week 3 ($p<0.0001$), reaching an apparent plateau at week 4. In Group M, cPass sVNT values increased moderately with a significant increase from week 2 to week 3 ($p<0.05$). The sVNT value of Group S was significantly higher than that of Group M during week 4. ($p<0.0001$, Mann–Whitney U test). Table 4 shows cPass sVNT positivity after symptom onset. In critical, severe, and mild cases, 100% sensitivity was observed 3 weeks after onset. Four weeks after symptom onset, all cases tested were positive for cPass sVNT.

sVN antibody values were measured for SARS-CoV-2 PCR-positive patient samples for the indicated weekly timeframes post-onset of symptoms using cPass sVNT. 84 samples from 20 severe to critical cases (Group S) and 80 samples from 45 mild to moderate cases (Group M) were tested. The levels of S-IgG antibody in Group S and Group M were compared. Gray bars indicate Group S and open bars indicate Group M. The vertical axis is in logarithmic notation. The data are presented as means with interquartile ranges. Statistical significance is indicated as follows: $^{*}p < 0.05$; $^{**}p < 0.0001$ (Mann-Whitney U test).

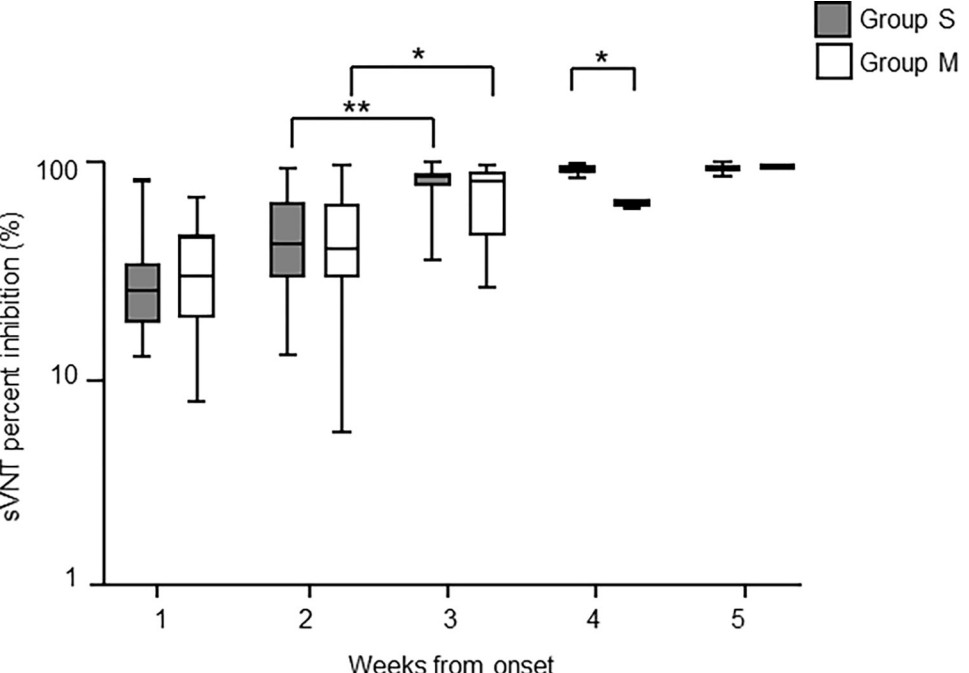

**Fig 3. The time course of cPass sVNT titers in COVID-19 patients after symptom onset.**

**Table 4. Clinical sensitivity of cPass sVNT.**

| weeks from onset | critical (n = 4)* | | severe (n = 16) | | moderate (n = 42) | | mild (n = 3) | |
|---|---|---|---|---|---|---|---|---|
| | sample # | positive # (%) | sample # | positive # (%) | sample # | positive # (%) | sample # | positive # (%) |
| 1 | 3 | 2 (67) | 6 | 2 (33) | 17 | 7 (41) | 1 | 0 (0) |
| 2 | 9 | 5 (56) | 31 | 22 (71) | 35 | 25 (71) | 4 | 2 (50) |
| 3 | 7 | 7 (100) | 18 | 18 (100) | 18 | 14 (78) | 1 | 1 (100) |
| 4 | 3 | 3 (100) | 5 | 5 (100) | 2 | 2 (100) | 1 | 1 (100) |
| 5 | 0 | 0 (N/A) | 2 | 2 (100) | 1 | 1 (100) | 0 | 0 (N/A) |

* patient number

N/A, not applicable

## Longitudinal assessment of antibody level in COVID-19 patients

To examine changes in antibody levels over time, we plotted the titers of inpatients measured two or more times in a row (Fig 4). A total of 190 samples from 46 cases were collected up to 31 days after symptom onset to determine the antibodies' rate of change. The 45 cases were divided into two groups: group S (20 cases, including 16 severe and 4 critical cases) and group M (25 cases, including 4 mild and 21 moderate cases). All mild, moderate, and severe cases

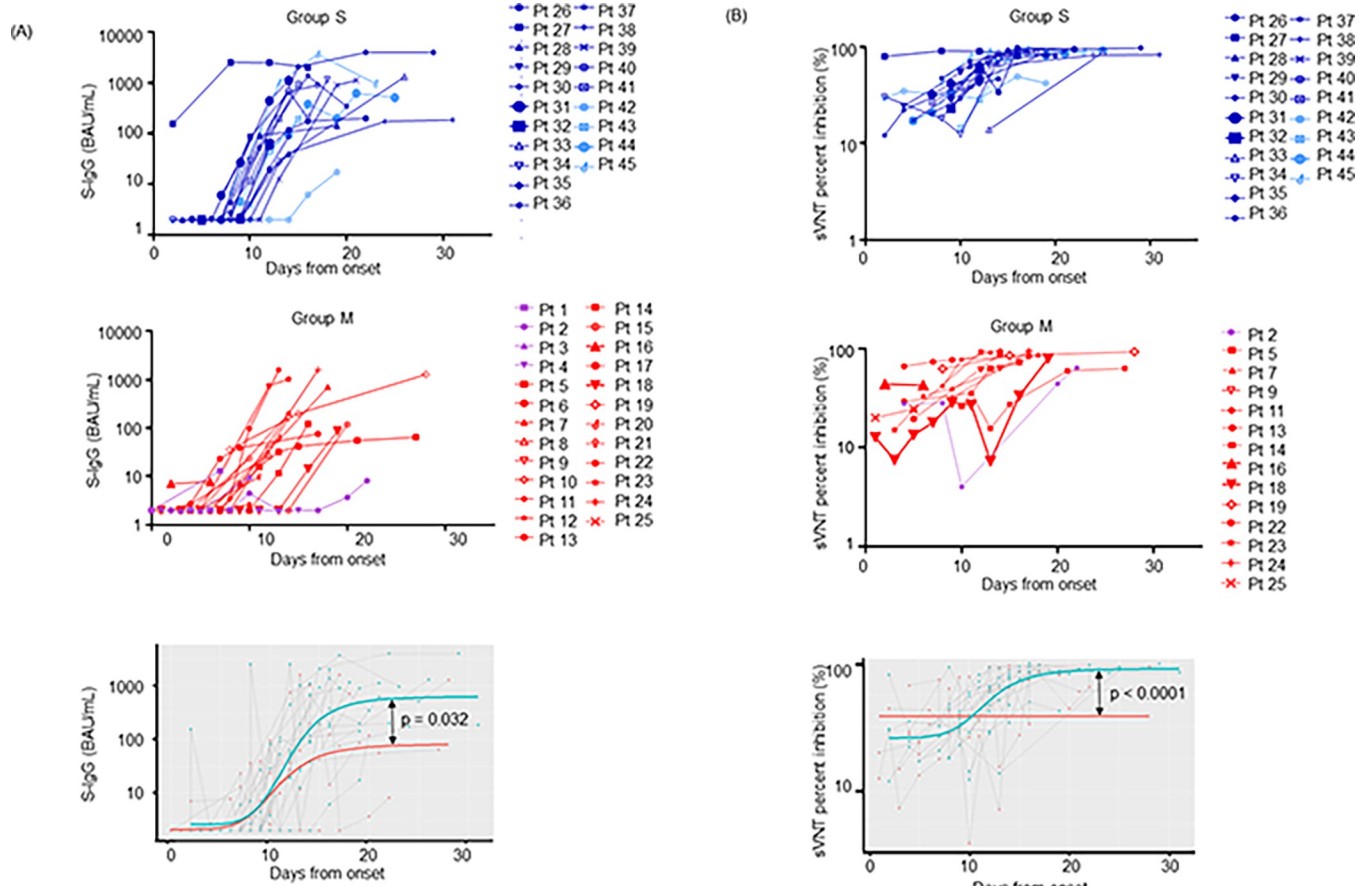

**Fig 4. Longitudinal change of S-IgG and sVNT values.**

were cured and discharged. All critical cases have deceased. We determined the kinetics of the emergence of S-IgG and NT antibodies using nonlinear mixed-effects models, as described in Materials and Methods. VITROS S-IgG values and cPass sVNT titers from hospitalized patients were plotted against time from symptom onset and fitted (Fig 4A and 4B, lower graphs). We observed highly significant differences of the plateau values between Group S and Group M individuals both for the VITORS S-IgG values and for the cPass sVNT titers ($p$ = 0.032 and $p<0.0001$ by Wilcoxon test, respectively).

Longitudinal changes of S-IgG antibody and sVN antibody levels were investigated for the indicated weekly timeframes post-onset of symptoms.

a. (A) The VITROS S-IgG assay was performed using 96 samples from 20 severe and critical cases (Group S) and 94 samples from 25 mild and moderate cases (Group M).

b. (B) The cPass sVNT was performed using 83 samples from 20 cases of Group S and 50 samples from 14 cases of Group M.

The graphs at the bottom show the comparisons of the fitted plateau values after day 15 post symptom onset of Group S and Group M for VITROS S-IgG (A) and cPass sVNT (B) titers (Wilcoxon Z test, lower panels). The vertical axes are in logarithmic notation. Group S: critical, light blue lines; severe, dark blue lines. Group M: moderate, red lines; mild, purple lines.

## Distribution of VITROS S-IgG and cPass sVNT values after second vaccination

Finally, we investigated VITROS S-IgG and cPass sVNT levels in 113 healthcare workers who received two doses of BNT162b2 mRNA vaccine by May 13, 2021. The serum samples were obtained between June 8 and 21. Because seropositive individuals with N-specific antibodies are considered previously infected with SARS-CoV-2 with asymptomatic COVID-19, the positivity of N-specific antibodies was further detected. All tested individuals, except one immunosuppressed case suffering from collagen disease, were seropositive with both VITROS S-IgG and cPass sVNT. The median antibody titer was 777.0 BAU/ml (IQR 457.7–1355.0) for S-IgG and 95.4% (IQR 89.8–97.1) for sVNT. The post-vaccination healthcare workers were then divided into 3 groups; Group 1, N-specific antibody-negative/no COVID-19 history (n = 73); Group 2, N-specific antibody-positive/no COVID-19 history (n = 25); and Group 3, with COVID-19 history, the time of onset of COVID-19 varied from 2 to 14 months (n = 15).

As shown in Fig 5, both VITROS S-IgG and cPass sVNT values were statistically lower in the individuals who were negative for N-specific antibody compared to the ones with a history of COVID-19 and those who were positive for N-specific antibodies without previously diagnosed COVID-19. The N-specific antibody positive individuals showed comparable S-IgG and sVNT titers to those of the ones who had been diagnosed with COVID-19. We further investigated whether the values of S-IgG and sVN antibody values were changed over time after vaccination. As shown in Fig 6, both VITROS S-IgG and cPass sVNT titers did not decrease significantly over time up to 75 days after the second vaccination, regardless of previous COVID-19 infection.

VITROS S-IgG levels (A) and cPass sVNT values (B) were quantified in post-vaccination healthcare workers (n = 113). Group 1, N-specific antibody negative without COVID-19 history (n = 73); Group 2, N-specific antibody positive without COVID-19 history (n = 25); Group 3, with COVID-19 history (n = 15). Open circles in Group 3 were the individuals with negative N-specific antibody (n = 3). The vertical axis of VITROS S-IgG levels (A) is

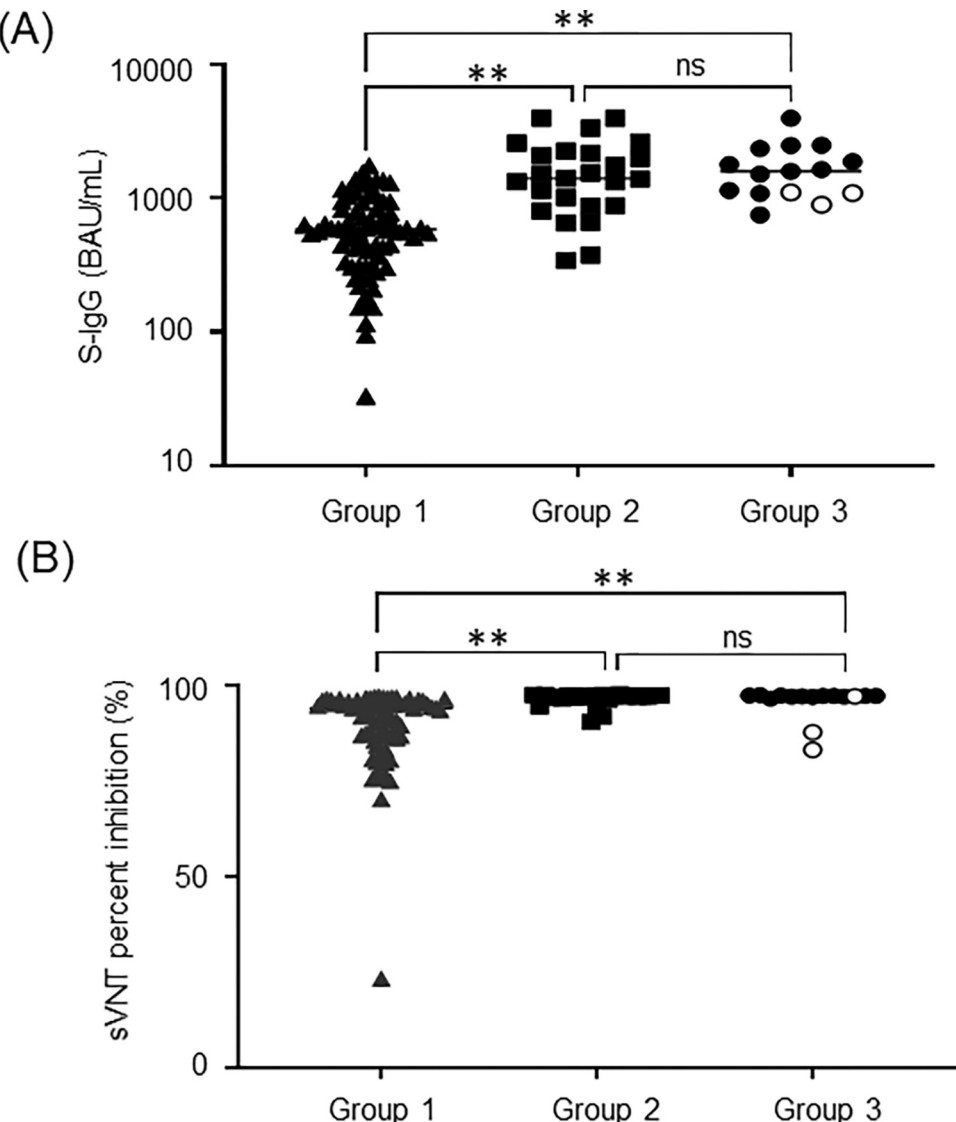

**Fig 5. Distribution of VITROS S-IgG and cPass sVNT values in participants after second vaccination.**

logarithmic notation. Statistical analysis was performed using one-way ANOVA, and statistical significance is indicated as follows: $**p < 0.0001$; ns, no significant difference.

VITROS S-IgG levels (A) and cPass sVNT values (B) were quantified in post-vaccination healthcare workers (n = 113). Group 1, N-specific antibody negative without COVID-19 history (n = 73); Group 2, N-specific antibody positive without COVID-19 history (n = 25); Group 3, with COVID-19 history (n = 15). Scatterplot and regression line colors indicate the antibody response. The 95% CIs are calculated by prediction ± 1.96 × standard error of prediction. The vertical axis of VITROS S-IgG levels (A) is in logarithmic notation.

## Discussion

In this study, we evaluated the commercially-available automated quantitative immunoassay Ortho VITROS SARS-CoV-2 Spike-Specific Quantitative IgG (VITROS S-IgG) test by comparing it with sVN antibody levels detected by the cPass sVNT and clinical assessment. To the

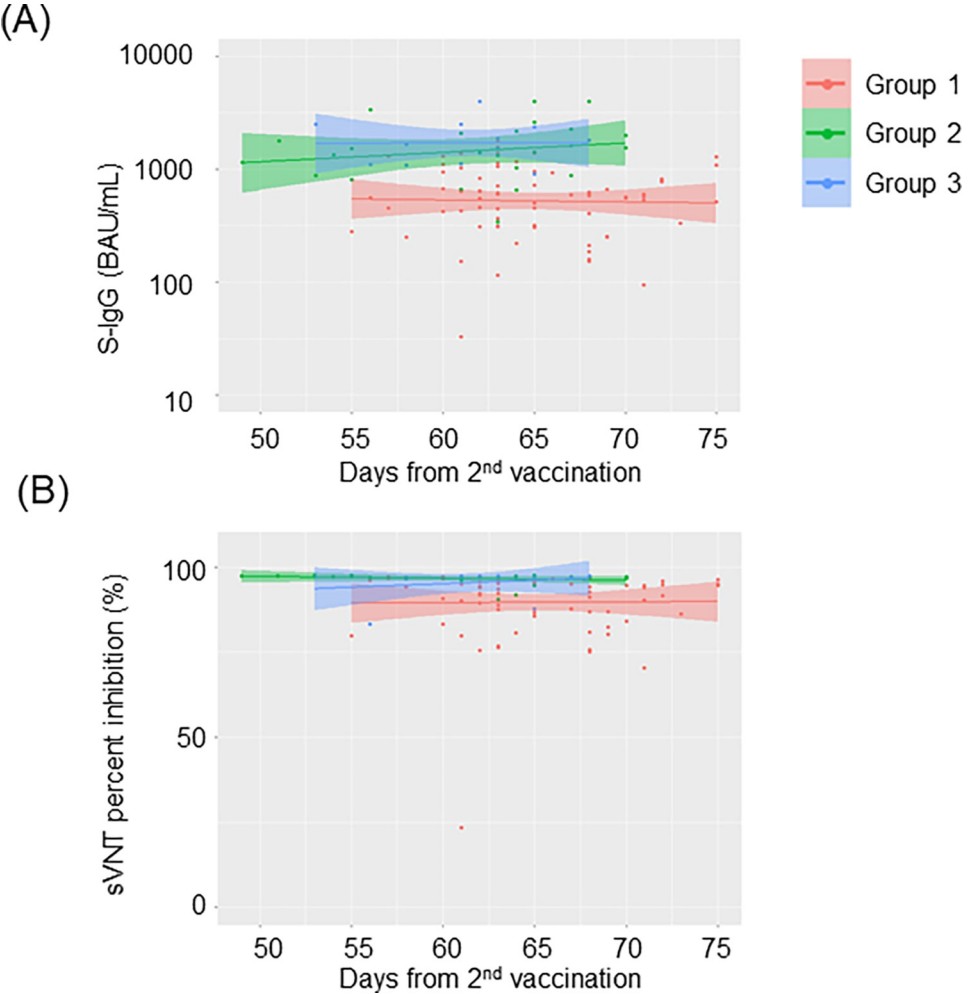

**Fig 6. Change in VITROS S-IgG and cPass sVNT titers over time after second vaccination.**

best of our knowledge, this is the first report to study the correlation of VITROS S-IgG with sVN antibodies.

Currently, the neutralizing activity of the detected S-specific antibodies after vaccination is a major concern. In response to this, sVNT was developed and reported to be correlated well with the "gold standard" plaque reduction neutralizing test (PRNT) [14, 15]. In this study, we observed that Ortho VITROS S-IgG immunoassay strongly correlated with the sVN antibody titers detected by cPass sVNT. These results consistent with recent reports concerning immunoassays other than VITROS S-IgG, which demonstrate good correlations between S-specific antibodies and NT antibodies measured by cPass sVNT [16, 17]. However, almost half of the VITROS S-IgG negative samples were found to be cPass sVNT positive. Moreover, in longitudinal evaluations from COVID-19 patients, S1-IgG was negative in all mild cases, but cPass sVNT was positive in some. VITROS S-IgG quantitatively detects only IgG subclass antibodies against the S1 subunit of the spike protein. In contrast, cPass sVNT qualitatively detects total surrogate neutralizing antibodies in an isotype-independent manner which determines antibodies have neutralizing activity (i.e., binding inhibitory effect) if they bind to RBD by 30% or more. Previous reports have shown that the sVNT assay detects a substantial level of sVN antibodies regardless of the IgM/IgG ratio [13], which indicates that there are sVN antibodies with

RBD binding ability even below the cutoff value of Vitros S-IgG. However, further research is warranted to determine whether sVNT detected NT antibody levels are directly related to protection against infection.

Two weeks after symptom onset, Group S showed significantly higher values than Group M in both VITROS S-IgG and cPass sVNT assays. These findings are consistent with previous reports demonstrating that elevated NT antibody levels due to SARS-CoV-2 coincide with disease progression [18, 19]. Because NT antibodies can block infection directly, the role of the antibody response in COVID-19 immunopathology is unclear.

In terms of the COVID-19 humoral immune response after vaccination, we observed good agreement between VITROS S-IgG and cPass sVNT levels in the healthcare workers sampled 2 months after the second dose of BNT162b2 vaccination. A high titer of S-specific antibodies was observed in N-specific seropositive individuals who have not been diagnosed with COVID-19 by RT-PCR since they lacked COVID-19 related symptoms. S-specific antibody titers of N-positive individuals were comparable to those of COVID-19 infected cases. We did not observe significant decrease of VITROS S-IgG and cPass sVNT titers up to 75 days after the second vaccination. Several studies on the durability of humoral response have shown that levels of both S-IgG and NT antibody decrease modestly until about 8 months after SARS-CoV-2 infection in recovered cases [20, 21]. However, significant reductions in these antibodies have been reported within 6 months after the second dose of the BNT162b2 vaccine [22], with frequent incidence of breakthrough infections [23, 24]. In this study, no significant decrease in VITROS S-IgG and cPass sVNT titers was observed up to 75 days after the second vaccination, regardless of previous COVID-19 infection history. More long-term longitudinal evaluations are required to clarify whether previous infections affect the efficacies of vaccinations.

This study had several limitations. First, it was conducted in a single university hospital. Second, specificity of the tests has not been validated using pre-COVID-19 clinical specimens. Third, because the COVID-19 samples were obtained from hospitalized patients after SARS-CoV-2 wave, specificity was not evaluated with the samples pre wave and asymptomatic COVID-19 cases were not included. Forth, post-vaccination antibody measurements were made only once and chronological changes in antibody titers could not be followed for the same individuals. Fifth, to measure NT antibody activity levels, we utilized the cPass sVNT kit, a surrogate test for NT antibody, but did not perform the "gold standard" PRNT.

Antibody responses represent key immune correlates of protection for SARS-CoV-2 as well as a diagnostic tool. VITROS automated quantitative immunoassay system offers high-throughput, widely available laboratory measurement of antibodies as an advantage compared to the time-consuming, low-throughput cPass sVNT. However, this study revealed that patients with low antibody titers, such as mild cases of COVID-19, could be cPass sVNT positive but VITROS S-IgG negative. This is a major disadvantage of VITROS S-IgG for use as a quantitative marker of neutralizing activity.

In conclusion, we observed that the automated quantitative immunoassay VITROS S-IgG showed good diagnostic performance and a reasonable correlation with the sVN antibodies detected by the cPass sVNT. However, this study also demonstrated the limitation in using VITROS S-IgG as a direct quantitative marker of neutralization activity capacity. These findings indicate that the VITROS S-IgG may be a useful diagnostic tool and can be utilized to assess response to vaccination and herd immunity with careful interpretation.

## Acknowledgments

The authors thank the Department of Research Support Utilizing Bioresource Bank, Juntendo University Graduate School of Medicine, for use of their facilities.

## Author Contributions

**Conceptualization:** Kaori Saito, Yoko Tabe.

**Data curation:** Takashi Miida.

**Formal analysis:** Maika Takahashi, Tomohiko Ai, Shuko Nojiri, Takamasa Yamamoto, Mitsuru Wakita, Yoko Tabe.

**Funding acquisition:** Toshio Naito.

**Investigation:** Maika Takahashi, Takamasa Yamamoto, Mitsuru Wakita.

**Methodology:** Maika Takahashi, Takamasa Yamamoto.

**Resources:** Makoto Hiki.

**Supervision:** Takashi Miida, Toshio Naito, Kazuhisa Takahashi, Yoko Tabe.

**Writing – original draft:** Maika Takahashi, Tomohiko Ai, Shuko Nojiri, Yoko Tabe.

**Writing – review & editing:** Abdullah Khasawneh, Faith Jessica Paran, Yuki Horiuchi, Satomi Takei.

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
