## [Decision Letter · Decision Letter 0]

8 Jul 2022

PONE-D-22-15685Performance evaluation of the Ortho VITROS SARS-CoV-2 Spike-Specific Quantitative IgG test by comparison with neutralizing antibody and clinical assessmentPLOS ONE

Dear Dr. Tabe,

Thank you for submitting your manuscript to PLOS ONE. After careful consideration, we feel that it has merit but does not fully meet PLOS ONE’s publication criteria as it currently stands. Therefore, we invite you to submit a revised version of the manuscript that addresses the points raised during the review process.

As pointed out by one of the reviewers, please describe the details of patient characters.

We look forward to receiving your revised manuscript.

Kind regards,

Etsuro Ito

Academic Editor

PLOS ONE

Journal Requirements:

"This work was supported in part by This research was partially supported by AMED under Grant Number JP20fk0108472 to TN."

"Ortho Clinical Diagnostics provided Anti-SARS-CoV-2 IgG Quantitative Reagent, and Roche Diagnosis provided reagents for Elecsys Anti-SARS-CoV-2 assay free of cost to the researchers. The companies did not take part in 1) the study design, 2) the data interpretation, and 3) the writing of this paper."

6. Please amend the manuscript submission data (via Edit Submission) to include author Takashi Miida.

Reviewers' comments:

Reviewer's Responses to Questions

**Comments to the Author**

1. Is the manuscript technically sound, and do the data support the conclusions?

Reviewer #1: Yes

Reviewer #2: Partly

2. Has the statistical analysis been performed appropriately and rigorously? 

Reviewer #1: Yes

Reviewer #2: Yes

3. Have the authors made all data underlying the findings in their manuscript fully available?

Reviewer #1: Yes

Reviewer #2: Yes

4. Is the manuscript presented in an intelligible fashion and written in standard English?

Reviewer #1: Yes

Reviewer #2: Yes

5. Review Comments to the Author

Reviewer #1: Takahashi M. et al. evaluated the performance of the Ortho VITROS SARS-CoV-2 Spike-Specific Quantitative IgG kit. This work would give the useful information to the readers, but it is necessary to improve this manuscript for publication in PLoS One.

1) Line 229: Table 1 does not summarize the clinical background characteristics. Please add the table to show the clinical information.

2) Fig. 1: The power of antibody detection seems different between cPass sVNT and VITROS S-IgG. Why? Please explain the reason.

3) Fig. 2: Fig. 2 (A), (B), and (C) are redundant. Please avoid the repetition. (I think that all results shown in (A), (B), and (C) can be unified in (C).)

4) Fig.3: Fig. 3 (A), (B), and (C) are also redundant. Please avoid the repetition.

5) Please show and discuss the specificity and sensitivity of VITROS S-IgG and cPass sVNT. Authors should clearly state the advantages and disadvantages of VITROS S-IgG as compared with cPass sVNT.

6) I recommend adding the new table for easy understanding of the details of VITROS S-IgG and cPass sVNT (manufacturer, method, antigen, immunoglobulin class, unit etc.).

7) The amount of antibody can be affected by how long time has passed after vaccination. However, there was no information on the timing of sample collection from the vaccinees. Please show this point.

8) In addition, no information was shown when the patients in Group 3 were vaccinated and diagnosed with COVID-19. The timings of vaccination, sample collection, and infection should influence the antibody titer. Please clarify these points.

Reviewer #2: In this manuscript, Takahashi et al evaluated the performance of Ortho VITROS SARS-CoV-2 spike-specific quantitative IgG (VITROS S-IgG) assay, in comparison with GenScript cPass SARS-CoV-2 Neutralization Antibody Detection Kit (cPass sVNT assay). They described that VITROS S-IgG showed good correlation with the cPass sVNT assay. They concluded that VITROS S-IgG is useful as a diagnostic tool and can be utilized for assessing immune response to vaccination and herd immunity.

This is the first report to show the correlation of VITROS S-IgG with NT antibodies, but there are several points which have to be improved for publication in PLoS One.

Major points

1. Page 7, lines 108 to 112.

The authors should describe the details of the patients’ characteristics of Group S and Group M, which are explained in the COVID-19 clinical management Living Guideline from WHO.

2. Page 6, lines 90 to 94.

cPass sVNT assay is a surrogate test for neutralizing antibody using pseudovirus as the authors mentioned in the abstract and the main text; the comparison shown in this manuscript was not performed using cell-culture test which is the gold standard method for virus-neutralization. The authors should describe it as a limitation of this study in the discussion.

3. Figure 1 and Table 1

There are few description regarding Table 1 in the main text. Assuming that Figure 1 and Table 1 show the same content, the correlation between VITROS S-IgG and cPass sVNT assay is not excellent; The agreement of result calculated from Table 1 is 85%. The authors should explain it in the discussion.

Minor points

1. Page 8, lines 116 to 118.

The approvable number from Juntendo IRB is required.

2. Page 14, line 229.

Table 1 shows “Agreement between Vitros S-IgG and cPass”, but not “summarizes the clinical background characteristics”. The authors should reconcile these expression. In addition, the order of items in the table 1 should be corrected; VITROS S-IgG should be shown in the left column, rather than in the upper right in the table. The explanation for “(%)” is needed, because it is hard to understand which ratio is shown as “(%)”. “Positive” should be shown preferentially upward than “Negative”.

6. PLOS authors have the option to publish the peer review history of their article (what does this mean?). If published, this will include your full peer review and any attached files.

Reviewer #1: No

Reviewer #2: No

---

## [Author Response · Author response to Decision Letter 0]

21 Sep 2022

Comments are addressed in the Response to Reviewers file

---

## [Decision Letter · Decision Letter 1]

20 Oct 2022

PONE-D-22-15685R1Performance evaluation of the Ortho VITROS SARS-CoV-2 Spike-Specific Quantitative IgG test by comparison with neutralizing antibody and clinical assessmentPLOS ONE

Dear Dr. Tabe,

Thank you for submitting your manuscript to PLOS ONE. After careful consideration, we feel that it has merit but does not fully meet PLOS ONE’s publication criteria as it currently stands. Therefore, we invite you to submit a revised version of the manuscript that addresses the points raised during the review process. After your revision, I sent your manuscript to the 2 reviewers who previously reviewed.One of the reviewers was satisfied with your revision but the other one was not (he decided reject).So, I had to send your manuscript to other two reviewers for the fair judgement and then received the comments from these two.Thus, I need to ask you to revise your manuscript again.Please consider the comments from Reviewers #3 and #4.Thank you for your patience.

We look forward to receiving your revised manuscript.

Kind regards,

Etsuro Ito

Academic Editor

PLOS ONE

Journal Requirements:

Reviewers' comments:

Reviewer's Responses to Questions

**Comments to the Author**

1. If the authors have adequately addressed your comments raised in a previous round of review and you feel that this manuscript is now acceptable for publication, you may indicate that here to bypass the “Comments to the Author” section, enter your conflict of interest statement in the “Confidential to Editor” section, and submit your "Accept" recommendation.

Reviewer #1: (No Response)

Reviewer #2: All comments have been addressed

Reviewer #3: (No Response)

Reviewer #4: All comments have been addressed

2. Is the manuscript technically sound, and do the data support the conclusions?

Reviewer #1: Partly

Reviewer #2: (No Response)

Reviewer #3: Yes

Reviewer #4: Yes

3. Has the statistical analysis been performed appropriately and rigorously? 

Reviewer #1: Yes

Reviewer #2: (No Response)

Reviewer #3: Yes

Reviewer #4: Yes

4. Have the authors made all data underlying the findings in their manuscript fully available?

Reviewer #1: Yes

Reviewer #2: (No Response)

Reviewer #3: Yes

Reviewer #4: Yes

5. Is the manuscript presented in an intelligible fashion and written in standard English?

Reviewer #1: Yes

Reviewer #2: (No Response)

Reviewer #3: No

Reviewer #4: Yes

6. Review Comments to the Author

Reviewer #1: I feel that the quality of the revised manuscript has been much improved, but this version still has some points to be changed for considering publication in PLoS One.

1) I would like to emphasize that cPass sVNT assay does not show NT titers but the inhibition values of ACE2-RBD interaction as the surrogate based on ELISA. The use of the word "neutralizing antibody" or "NT" would cause the readers to misunderstand. I recommend using other words like "surrogate neutralizing antibody" or "sVNT" in the whole manuscript. Please consider the revision of the title, in particular.

2) The authors concluded that VITROS S-IgG shows good correlation with the cPass sVNT assay. However, I feel that the interpretation of the results by the authors was not reasonable. Table 2 demonstrates that the performance of VITROS S-IgG was quite different from that of cPass sVNT. Table 3 and Table 4 also show that cPass sVNT had the higher clinical sensitivity than the VITROS S-IgG (for example, cPass sVNT assay showed 4 positives in 7 samples, but VITROS S-IgG assay did 0 positive in 20 samples in mild cases). Fig.1 to 4 clearly exhibited that VITROS S-IgG had the lower sensitivity than cPass sVNT.

3) Previously, I requested the authors to explain the reason why the power of antibody detection was higher in cPass sVNT than VITROS S-IgG. However, the explanation in the revised manuscript was still insufficient.

Reviewer #2: (No Response)

Reviewer #3: The authors compared the clinical performance of VITRO S-IgG and the NT antibody levels (sVNT assay) using 332 serum samples collected from 188 individuals. The samples were grouped into Group S (severe or critical) and Group M (mild or moderate) with 113 samples obtained from healthcare workers who had received two doses of BNT162b2 vaccine.

The manuscript is well written, and the methodology of the research and its statistical analysis is in a sound manner. In addition, the authors adequately have followed the suggestions of the reviewers and revised their manuscript appropriately. Although these may be trivial things, the manuscript seems to contain some erroneous grammatical usages of English. They should be amended before the manuscript reaches the decision of acceptance for its publication in PLoS One. It is strongly recommended that the manuscript should undergo English editing services. Furthermore, there is a minor point which should be considered for the authors to revise, as below.

A minor point: in line 310(R1), table 5 is not informative and should be omitted. This table only presents the number of the patients accompanied by their disease severity, group, sex, and past medical history, all of which are not relevant to the substance of this research. This information can be summarized and simply included in the main text.

Reviewer #4: The manuscript is well written but a few changes are recommended. IN the abstract , this sentence is unclear and (N-specific) is not defined.

"In regard to the COVID-19 humoral immune response after the second dose of the BNT162b2 vaccination, similar levels of VITROS S-IgG and cPass sVNT were observed with high titers in N-specific seropositive individuals in both VITROS S-IgG and cPass sVNT."

In the discussion (line 391) the is a paragraph about complement, T cells. This manuscript look at neither complement nor T cells and the paragraph seems speculative and not based on the current data.

Table 4 is cutoff on right side

There are 2 blank pages.

7. PLOS authors have the option to publish the peer review history of their article (what does this mean?). If published, this will include your full peer review and any attached files.

Reviewer #1: No

Reviewer #2: No

Reviewer #3: No

Reviewer #4: No

---

## [Author Response · Author response to Decision Letter 1]

30 Nov 2022

The comments were addressed and summarized in a separate word file, "Response to Reviewers"

---

## [Decision Letter · Decision Letter 2]

14 Dec 2022

Performance evaluation of the Ortho VITROS SARS-CoV-2 Spike-Specific Quantitative IgG test by comparison with the surrogate virus neutralizing antibody test and clinical assessment

PONE-D-22-15685R2

Dear Dr. Tabe,

We’re pleased to inform you that your manuscript has been judged scientifically suitable for publication and will be formally accepted for publication once it meets all outstanding technical requirements.

Kind regards,

Etsuro Ito

Academic Editor

PLOS ONE

Reviewers' comments:

Reviewer's Responses to Questions

**Comments to the Author**

1. If the authors have adequately addressed your comments raised in a previous round of review and you feel that this manuscript is now acceptable for publication, you may indicate that here to bypass the “Comments to the Author” section, enter your conflict of interest statement in the “Confidential to Editor” section, and submit your "Accept" recommendation.

Reviewer #3: (No Response)

Reviewer #4: All comments have been addressed

2. Is the manuscript technically sound, and do the data support the conclusions?

Reviewer #3: (No Response)

Reviewer #4: Yes

3. Has the statistical analysis been performed appropriately and rigorously? 

Reviewer #3: (No Response)

Reviewer #4: Yes

4. Have the authors made all data underlying the findings in their manuscript fully available?

Reviewer #3: (No Response)

Reviewer #4: (No Response)

5. Is the manuscript presented in an intelligible fashion and written in standard English?

Reviewer #3: (No Response)

Reviewer #4: Yes

6. Review Comments to the Author

Reviewer #3: The authors addressed all the comments raised by the reviewer. This research is relatively small-scale but performed in a sound manner.

Reviewer #4: Looks ok for publication . All comments have been addressed . The authors have responded to criticism and made changes as suggested

7. PLOS authors have the option to publish the peer review history of their article (what does this mean?). If published, this will include your full peer review and any attached files.

Reviewer #3: No

Reviewer #4: No

---

## [Editor Report · Acceptance letter]

16 Jan 2023

PONE-D-22-15685R2 

Performance evaluation of the Ortho VITROS SARS-CoV-2 Spike-Specific Quantitative IgG test by comparison with the surrogate virus neutralizing antibody test and clinical assessment 

Dear Dr. Tabe:

I'm pleased to inform you that your manuscript has been deemed suitable for publication in PLOS ONE. Congratulations! Your manuscript is now with our production department. 

Kind regards, 

on behalf of

Prof. Etsuro Ito 

Academic Editor

PLOS ONE